# AGENTDISTILL: TRAINING-FREE AGENT DISTILLATION WITH GENERALIZABLE MCP BOXES

## ABSTRACT

While knowledge distillation has become a mature field for compressing large language models (LLMs) into smaller ones by aligning their outputs or internal representations, the distillation of LLM-based agents, which involve planning, memory, and tool use, remains relatively underexplored. Existing agent distillation methods typically replay full teacher trajectories or imitate step-by-step teacher tool usage, but they often struggle to train student agents to dynamically plan and act in novel environments. We propose **AgentDistill**, a novel, training-free agent distillation framework that enables efficient and scalable knowledge transfer via direct reusage of Model–Context–Protocols (MCPs)—structured and reusable task-solving modules autonomously generated by teacher agents. The reuse of these distilled MCPs enables student agents to generalize their capabilities across domains and solve new problems with minimal supervision or human intervention. Experiments on biomedical and mathematical benchmarks demonstrate that our distilled student agents with small language models can achieve performance comparable to advanced systems with strong LLMs such as OctoTools (GPT-4o). The distilled agents achieve 58.9% average accuracy on biomedical VQA benchmarks (vs. 58.5% with OctoTools) and 75.5% on the mathematical Game of 24 benchmark (vs. 45% with OctoTools). These results highlight the effectiveness of our framework in building scalable, cost-efficient, and generalizable LLM-based agents.

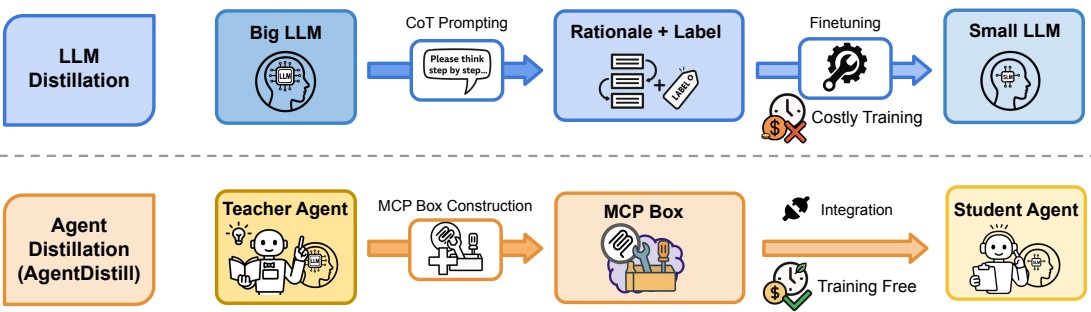

Figure 1: Comparison between traditional LLM distillation (top) and our proposed training-free agent distillation framework (bottom). Traditional LLM distillation relies on chain-of-thought prompting followed by costly fine-tuning on rationale–label pairs, whereas our method eliminates training entirely. Instead, a teacher agent autonomously generates modular and reusable Model–Context–Protocols (MCPs), which are directly integrated into student agents. This enables sLM-based agents to inherit task-solving capabilities without gradient updates or trajectory replay.

## 1 INTRODUCTION

Large language model (LLM) distillation has become a widely used technique to reduce inference cost while retaining most teacher performance. Early knowledge distillation (KD) methods align student and teacher output logits (Hinton et al., 2015; Sanh et al., 2019). Later work shows that matching hidden features (Sun et al., 2019; Jiao et al., 2019), attention patterns (Wang et al., 2020), and using architecture-aware objectives (Sun et al., 2020; Tan et al., 2023) can further close the performance gap between the student and teacher model. Chain-of-thought distillation (CoTD) teaches students to follow step-by-step rationales generated by teachers (Ho et al., 2022; Shridhar et al., 2023), sometimes using sampled or structured traces to highlight the critical steps (Li et al., 2023; 2022; Feng et al., 2024).

Beyond language models, recent efforts have begun to explore how distillation techniques can be extended to LLM-based agents that integrate reasoning with tool use and environment interaction. These efforts vary widely in how they conceptualize agent behavior and what aspect of the teacher they aim to transfer. One type of work trains student agents to imitate reasoning-action trajectories from teacher agents, such as Structured Agent Distillation (SAD) (Liu et al., 2025) and retrieval-augmented distillation methods (Kang et al., 2025). These methods treat agent behavior as interleaved thoughts and tool calls, supervising the student to mimic each step. While effective in capturing execution details, they incur high computational cost and generalize poorly, as teachers require constructing and processing long, complex sequences, and students passively replicate fixed trajectories without learning to adapt. For works in structure distillation, like MAGDi (Chen et al., 2024a) and Sub-goal Distillation (Hashemzadeh et al., 2024), although they are more efficient than trajectory distillation, guiding students with abstracted teacher strategies like subgoal sequences or interaction graphs, these methods overlook differences in model capability, knowledge boundaries, or tool usage between different models.

To address the limitations of trajectory imitation and structured plan distillation—namely high computational cost and limited adaptability—we propose a lightweight, training-free framework: **AgentDistill**. Rather than replicating full trajectories or assuming students can execute teacher-defined plans, our approach leverages the inherent strengths of teacher agents in coding and task-solving by utilizing teacher-generated Model–Context–Protocols (MCPs) [1]. MCP is an open protocol designed to standardize how context is provided to LLMs. Our framework capitalizes on the teacher agent's capacity to create self-contained, reusable, and generalizable MCPs tailored to specific task domains. These MCPs encapsulate the problem-solving capabilities of the teacher agent and enable student agents equipped with substantially smaller LLMs (e.g., llama-3.1-8B, Qwen3-8B) to inherit sophisticated, transferable problem-solving skills without additional training. By directly integrating these distilled MCP boxes, student agents significantly enhance their performance and adaptability, effectively bridging the capability gap between teacher and student agents. Consequently, our method offers a scalable, efficient, and low-cost solution for agent distillation, enabling student agents to robustly handle diverse real-world scenarios.

We conduct comprehensive experiments on several benchmarks, including biomedical and mathematical tasks, to evaluate the effectiveness of our proposed AgentDistill framework across different domains. These results demonstrate that our approach substantially enhances the adaptability and generalization performance of student agents across diverse settings covered by teacher-generated MCPs, while also reducing inference and training costs. To summarize, our key contributions can be highlighted as follows:

- We propose AgentDistill, a novel agent distillation framework that enables student agents to inherit the more modular, transferable, and interpretable components—Model–Context–Protocols (MCPs)—generated by teacher agents. Unlike prior methods that rely on replaying long sequences of actions generated by the teacher, this approach allows student agents to directly inherit task-solving capabilities from teachers.

- AgentDistill is entirely a training-free framework. It requires no fine-tuning of either the teacher or the student agent. MCPs are automatically extracted, abstracted, and reused without additional gradient updates or handcrafted tool usage. This yields a highly cost-efficient and deployable distillation pipeline with strong generalization performance of the student to unseen tasks that can be solved with the distilled MCPs.

- We demonstrate that AgentDistill significantly enhances the problem-solving and generalization performance of student agents on biomedical and mathematical reasoning tasks, effectively narrowing the gap between teacher and student agents with minimal computational overhead. The comprehensive experiments are conducted across biomedical (PathVQA, SLAKE) and mathematical (Game of 24) benchmarks. Our proposed MCP distillation improves performance across all student models—`GPT-3.5-turbo`, `Qwen3-8B`, and `LLaMA3.1-8B`—with detailed gains shown in Table 2.

## 2 RELATED WORKS

### 2.1 DISTILLATION OF LARGE LANGUAGE MODEL

**Knowledge Distillation.** Knowledge distillation (KD) transfers knowledge from a large teacher model to a smaller student model by using teacher-provided soft targets and/or hidden representations. Early methods focus on aligning output probability distributions (Hinton et al., 2015; Sanh et al., 2019). Intermediate-layer feature alignment is used in patient distillation and two-stage distillation frameworks (Sun et al., 2019; Jiao et al., 2019). Self-attention matrix distillation captures internal Transformer relationships (Wang et al., 2020). Architecturally aware techniques modify network structures and perform joint distillation, as in MobileBERT and GKD (Sun et al., 2020; Tan et al., 2023).

---

[1] https://www.anthropic.com/news/model-context-protocol

Recent cross-model capability distillation uses large LLM–generated instruction–response pairs to teach smaller open models reasoning skills (Taori et al., 2023; Mukherjee et al., 2023).

**Reasoning Distillation.**    Chain-of-thought distillation (CoTD) methods train a smaller student model to reproduce a teacher's step-by-step reasoning via teacher-generated rationales and answers. Some approaches fine-tune students on full reasoning chains (Ho et al., 2022; Shridhar et al., 2023; Yang et al., 2025b) or on structured/sampled rationales (Li et al., 2023; 2022), ensuring students learn key reasoning patterns even with limited data. Other techniques focus training on critical steps or enforce faithfulness by sampling/weighting important tokens (Feng et al., 2024), maximizing mutual information (Chen et al., 2024b), or using contrastive decoding (Wang et al., 2023). To preserve core reasoning signals, long chains can be split into shorter chunks (Chen et al., 2025; Yang et al., 2025a), or aligned to alternative formats like trees or graphs (Zhuang et al., 2025). Finally, counterfactual distillation improves causal robustness (Chen et al., 2022), and domain-specialized distillation concentrates on task-specific CoT paths to boost performance on targeted benchmarks (Fu et al., 2023).

**In-Context Learning Distillation.**    In-context learning distillation (ICLD) (Snell et al., 2022; Upadhayayaya et al., 2024; Huang et al., 2022; Duan et al., 2024) trains a smaller student model to internalize a teacher's few-shot reasoning without requiring full prompts at inference. This has proven effective on benchmarks like NLI and SQL and is now standard in post-training. To enhance robustness, recent work integrates token-level language-modeling objectives (Huang et al., 2022) or treats few-shot matching as the sole training target (Duan et al., 2024), guiding students to internalize reasoning patterns.

## 2.2   Distillation of LLM Agent

**Trajectory Distillation.**    Trajectory-level agent distillation trains small models to imitate complete reasoning-action trajectories from large LLM-based agents. Structured Agent Distillation (SAD) (Liu et al., 2025) segments trajectories into interleaved thought and action spans, training students to reproduce agent-style execution patterns. Distilling LLM Agents into Small Models (Kang et al., 2025) extends this by including retrieved evidence and code execution results, enabling small models to emulate tool-augmented reasoning. These methods extend CoT distillation to agent settings by preserving not only intermediate reasoning but also tool usage and task decomposition behaviors.

**Structure Distillation.**    Structure-level agent distillation compresses reasoning trajectories into abstract representations such as graphs or subgoal sequences, enabling student models to preserve key task structures without imitating every token. MAGDi (Chen et al., 2024a) encodes multi-agent chats as interaction graphs, allowing students language model to reason over graph structure instead of raw text. Sub-goal Distillation (Hashemzadeh et al., 2024) extracts high-level goals from teacher agent trajectories and trains a student agent to predict and carry out the task plan. These methods reduce sequence length while preserving key reasoning patterns.

**Action Policy Distillation.**    Action policy distillation transfers language-based reasoning from LLM agents to lightweight, non-linguistic controllers. The teacher generates chain-of-thought trajectories in natural language, while the student executes actions directly without text generation. In Language-Oriented to Emergent Communication (Kim et al., 2024), a language agent trains an emergent-signal policy that communicates via short learned symbols. DeDer (Choi et al., 2024) converts reasoning traces into state-action pairs to train a small embodied agent for language-free execution.

Other training-free agent distillation methods, such as Agents Help Agents(Li et al.), distill by storing solved examples into a memory for retrieval-based in-context learning, which limits transfer to specific exemplars and does not endow the student with reusable task-solving procedures.

## 3   Methodology

To bridge the capability gap between a teacher agent leveraging large language models (LLMs), such as Claude-sonnet-4 or GPT-4o, and a student agent employing significantly smaller models (e.g., llama-3.1-8B, Qwen3-8B), we introduce a novel agent distillation framework called **AgentDistill**. The core concept behind AgentDistill is straightforward yet powerful: the teacher agent generates self-contained MCPs during task execution. These MCPs subsequently undergo a process of MCP box construction with abstraction, clustering, and consolidation, resulting in a MCP box that are then integrated into the student agents. This structured distillation process facilitates the transfer and internalization of sophisticated problem-solving skills initially demonstrated by the teacher agent, thereby substantially enhancing the capabilities of the student agent.

Figure 2: Overview of **AgentDistill**, a training-free agent distillation framework via Model–Context–Protocols (MCPs). The teacher agent with large language model solves tasks by decomposing them through a Manager Agent and generating task-specific MCPs via open-source search, script generation, and virtual execution. Valid MCPs are abstracted, clustered, and consolidated into a reusable MCP Box. At inference, the student agent with a small language model leverages this MCP Box to perform tool-based reasoning without any fine-tuning or trajectory replay. This enables lightweight agents to inherit task-solving capabilities from stronger models efficiently.

## 3.1 PROBLEM FORMULATION

Given supervision pairs $\mathcal{D} = \{(x_i, y_i)\}_{i=1}^N$ and a teacher agent $\pi_T$, we aim to distill teacher-agent-generated MCPs to a self-contained MCP Box, thus to improve a student agent $\pi_S$ with small language model performance by supplying the MCP Box. No further gradient update is applied to student agent $\pi : \nabla_\theta \pi_S = 0$. Formally, we define the optimization problem as: $\max_{\mathcal{B} \subset \mathcal{L}} \mathbb{E}_{(x,y) \sim \mathcal{D}} \left[ \mathbb{I} \{\pi_S(x; \mathcal{B}) = y\} \right]$, where $\mathcal{L}$ denotes the space of all teacher-agent-generated MCPs, $\mathcal{B}$ is the MCP Box distilled from $\mathcal{L}$, and $\pi_S(x; \mathcal{B})$ represents the behavior of the student agent when given input $x$ augmented with guidance from the MCP Box. The indicator function $\mathbb{I}\{\cdot\}$ evaluates to 1 if the student's output matches the ground truth.

## 3.2 AGENTDISTILL PIPELINE

### 3.2.1 MCP CREATION

When solving an input $x_i \in \mathcal{D}$, the teacher agent $\pi_T$ interacts with an environment $\mathcal{E}$, producing a full reasoning trajectory: $\tau_i = (r_1, a_1, o_1, \ldots, r_{L_i}, a_{L_i}, o_{L_i})$, where $r_t \in R$ are reasoning tokens, $a_t \in A$ are action tokens (e.g., tool calls, MCP generation), and $o_t \in O$ are observations from the environment.

To better distinguish MCP scripts from the reasoning, we prompt the teacher agent to generate and separate structured, self-contained MCPs during its reasoning process. Within the trajectory $\tau_i$, the teacher may produce one or more MCPs corresponding to distinct subtasks. For each input example $x_i \in \mathcal{D}$, if the teacher agent generates a MCP at the $j$-th step of its trajectory, we denote this MCP as $\text{MCP}_{i,j} \in \mathcal{L}$. where $\mathcal{L}$ is the space of all extracted MCPs across the specific dataset. Each trajectory may yield multiple MCPs depending on the number of tool-related planning steps.

Only trajectories where $\pi_T(x_i) = y_i$ (i.e., successful completions) are considered for distillation. We collect $\text{MCP}_{i,j}$ into a temporary pool if the MCP snippet is syntactically correct and executable. The result is a large pool $\mathcal{L} = \{\text{MCP}_{i,j}\}$, which captures a rich but noisy set of tool-use strategies emitted by the teacher agent. These MCPs will then be processed into a compact and organized set $\mathcal{B}$, termed the MCP Box, via abstraction, clustering, and consolidation, as detailed in the next section 3.2.2.

### 3.2.2 MCP BOX CONSTRUCTION

After collecting all MCPs generated from successful teacher trajectories, we pass them to a high-capacity instruction-tuned LLM (e.g., Claude-Sonnet-4) to form a compact and structured repository called the **MCP Box**. This process proceeds in three steps.

**(1) Abstraction.** For each tool-related MCP segment extracted from correct teacher trajectories, we extract the relevant Python code and prompt the LLM to rewrite it into a reusable and parameterized format, i.e. each raw MCP $\text{MCP}_{i,j}$ is rewritten into a concise, task-agnostic form using prompt-based transformation:

$$\hat{\text{MCP}}_{i,j} = \text{LLM}_{\text{abstract}}(\text{MCP}_{i,j}). \tag{1}$$

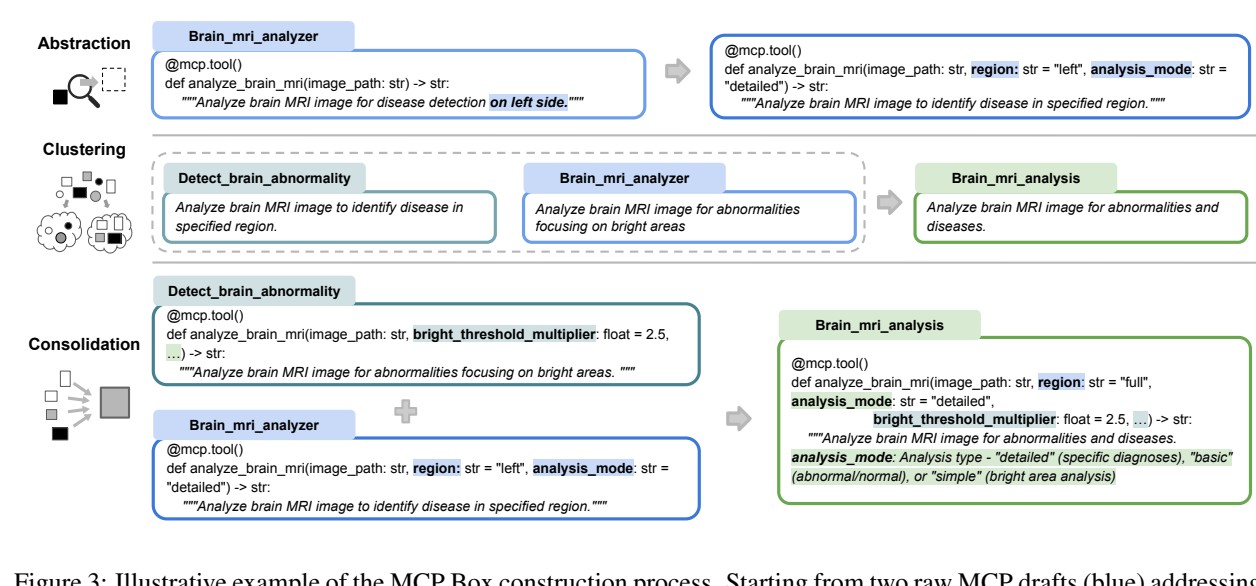

Figure 3: Illustrative example of the MCP Box construction process. Starting from two raw MCP drafts (blue) addressing related subtasks with overlapping functionality but differing parameters, we apply (1) abstraction to rewrite them into parameterized and reusable forms, (2) clustering to group functionally similar MCPs, and (3) consolidation to merge them into a single, general-purpose MCP (green) with configurable parameters. The resulting tool integrates multiple behaviors and is compatible with FastMCP execution.

The goal is to remove example-specific phrases while preserving generalizable tool-use strategies. Meanwhile, this process makes up to three critical parameters configurable, while preserving the tool's core logic.

**(2) Clustering.** All abstracted $\hat{\text{MCP}}_{i,j}$ are grouped by functionality via a code-level clustering prompt. The LLM returns cluster assignments based on shared application semantics:

$$\mathcal{C} = \text{LLM}_{\text{cluster}}\left(\left\{\hat{\text{MCP}}_{i,j}\right\}\right), \tag{2}$$

where each cluster $\mathcal{C}_k$ corresponds to a functional group like "image utils" or "numeric analysis".

**(3) Consolidation.** Within each cluster $\mathcal{C}_k$, we instruct the LLM to consolidate all tool implementations into a single general-purpose version. The result is

$$\text{MCP}_k^{\text{final}} = \text{LLM}_{\text{consolidate}}\left(\left(\{\hat{\text{MCP}}_{i,j} \mid \hat{\text{MCP}}_{i,j} \in \mathcal{C}_k\}\right)\right), \tag{3}$$

which includes parameter unification, proper validation, and documentation. Each output is a production-ready, FastMCP-compatible Python file. The complete MCP Box is then defined as $\mathcal{B} = \left\{(\text{MCP}_k^{\text{final}}, \text{cluster\_name}_k)\right\}_{k=1}^{K}$, where each item contains a consolidated tool protocol and its functional label.

### 3.2.3 STUDENT INFERENCE WITH THE MCP BOX

Based on the SmolAgents framework (Roucher et al., 2025), we mount the entire MCP-Box $\mathcal{B}$ into the student agent's tool interface at inference time—without retrieval, reranking, or parameter selection. Each $\text{MCP}_k^{\text{final}} \in \mathcal{B}$ is implemented as a callable tool with a standardized input/output interface (e.g., using @mcp.tool() within the FastMCP runtime). The student agent $\pi_S$ operates under a frozen policy and receives no gradient updates: $\nabla_\theta \pi_S = 0$.

When facing a new problem $x$, the student generates intermediate reasoning steps and tool calls as usual. At each step, the runtime environment exposes all tools in $\mathcal{B}$ as callable modules. The agent decides which tool to invoke (if any), fills in the input arguments (either through text generation or function call templates), and receives a return value $o_t$, which updates the context for the next reasoning step. No external scoring, selection, or retrieval is required. All tool-use competence is embedded in the preconstructed MCP-Box, allowing the student agent to benefit from distilled teacher knowledge with zero additional training. This design keeps the student agent lightweight and inference-time-efficient, while transferring all tool-related task-solving capability into the tool library itself.

### 3.3 Agent Structure

**Teacher Agent**    The teacher agent employs powerful large-scale language models (LLMs), renowned for their strong capabilities in coding and complex task-solving. To maintain simplicity and maximize efficiency, the teacher agent is designed with only three primary modules: a Manager Agent, a Basic Image Captioner, and an MCP Creation Module.

(1) Manager Agent. The Manager Agent serves as the central coordinator. Upon receiving a task prompt, the Manager Agent decomposes the task into manageable subtasks and evaluates whether external tools are required for their resolution. If external tools are necessary, it delegates the creation of Model–Context–Protocols (MCPs) to the MCP Creation Module. Following the execution of subtasks, the Manager Agent aggregates all intermediate results, synthesizing them into a coherent final response.

(2) Basic Image Captioner. This provides a textual summary of visual content when the input includes images. This component is especially important because many text-only models used do not support direct image input. The captioner converts images into textual descriptions, allowing the rest of the system, including the Manager and MCP Creation Module, to process visual information through a uniform text-based interface.

(3) MCP Creation Module. This module consists of four distinct sections: the MCP Brainstorming Section, the Open-Source Searching Section, the Script Generation Section, and the Virtual Environment Execution Section. The MCP Brainstorming Section generates initial conceptual plans for task-specific MCPs. Subsequently, the Open-Source Searching Section identifies relevant open-source resources to support MCP development. The Script Generation Section then synthesizes these ideas and resources into executable scripts. Finally, the Virtual Environment Execution Section validates and executes these scripts within a controlled environment, ensuring their practical applicability and robustness.

**Student Agent**    The student agent utilizes compact, cost-effective language models (e.g., llama-3.1-8B, Qwen3-8B) to significantly reduce inference expenses. Its structure closely mirrors that of the teacher agent but with a more streamlined composition, comprising only the Manager Agent and the Basic Image Captioner. The Manager Agent coordinates task decomposition, tool utilization, and result aggregation, benefiting directly from the distilled MCP box provided by the teacher agent, enabling it to efficiently handle complex tasks despite its smaller model scale.

## 4 Experiment

### 4.1 Experimental Setups

**Tasks and Datasets.**    We evaluate the effectiveness of AgentDistill in enhancing small language models (sLMs) on visual question answering (VQA) and mathematical tasks benchmarks. Specifically, we use **Game of 24** (Lile, 2025) for mathematical tasks and two real-world VQA datasets, **PathVQA** (He et al., 2020) and **SLAKE**(Liu et al., 2021). These datasets represent complex multi-hop reasoning over image-text pairs and require factual, visual inference capabilities, and precise symbolic arithmetic under strict constraints, enabling a comprehensive evaluation of agents' multi-modal and mathematical capabilities.

(1) Game of 24. The Game of 24 dataset is a mathematical benchmark with 1,362 puzzles. Each puzzle consists of four numbers to be combined using basic arithmetic operations to reach 24. Problems are ranked by human solving difficulty and include at least one valid solution.

(2) PathVQA. PathVQA is a pathology-focused visual question answering dataset containing 32,000 questions over 4,998 medical images. It emphasizes fine-grained visual reasoning in histopathology, such as identifying cell types or diagnostic markers.

(3)SLAKE. SLAKE is a multimodal medical VQA dataset with 642 radiology images and over 14,000 expert-annotated QA pairs. It tests both visual understanding and medical knowledge retrieval in a bilingual setting.

For each dataset, we sample 100 examples from validation set for MCP Box generation, same as benchmark dataset construction introduced in Octotools(Lu et al., 2025), and evaluate the student agent before distillation (without MCP box integration), after distillation (with MCP box integration), Student Agent with pre-defined tools (Octotools Framework), and the teacher agent on the same dataset. The results are summarized in Table 2.

**Models.**    Our experiments involve three small instruction-tuned language models (sLMs)—GPT-3.5-turbo, Qwen-8B, and LLaMA3.1-8B—which serve as the base of student agents in our study. We also use a teacher agent in which the Manager Agent is powered by Claude-Sonnet-4 and the MCP Creation Module is handled by

GPT-4o, representing an upper-bound reference. All models operate in a frozen configuration, without any task-specific fine-tuning or gradient updates.

**Settings.** We compare four settings: (1) student agents before distillation (without MCP Box); (2) agents with pre-defined tools (using Octotools Framework (Lu et al., 2025) and corresponding tools for each task) (3) student agents after distillation (with access to the distilled MCP Box); (4) the teacher agent; and (5) agents built upon OctoTools Framework engined by GPT-4o. This enables a comprehensive analysis of whether MCP narrows the gap between student agents and high-performance systems (either teacher agent or tool-augmented methods). See Figure 4 for cross-agent comparisons.

**Metrics.** We use task accuracy as the main evaluation metric, defined as the percentage of correctly answered dataset questions. To evaluate the benefit of MCP, we report the absolute improvement over the baseline sLMs before distillation. We also compare each student agent's performance with the teacher agent to assess whether distillation allows student agents to approach teacher agent performance.

## 4.2 RESULTS

We evaluate our approach across three datasets—PathVQA, SLAKE, and Game of 24—using multiple small language model (sLM) agents under the SmolAgent framework. All agents operate under a frozen policy and are equipped with the distilled **MCP Box** described in Section 3.

**Generalizability and Usage Frequency of Distilled MCPs.** Table 1 presents the number of unique MCPs generated by the teacher agent and the frequency with which student agents invoke them during inference. A high MCP Box calling rate indicates that distilled MCPs are broadly applicable across diverse inputs and consistently reused by student agents. These results confirm that our framework produces reusable and transferable MCPs that generalize well without requiring any additional training.

**MCP Box consistently improves student agents across datasets.** Table 2 shows that applying MCP leads to substantial improvements across all student agents and datasets. On PathVQA, GPT-3.5-turbo improves from 45.7% to 52.7%, Qwen-8B improves from 53% to 55.3%, and LLaMA3.1-8B improves from 46.7% to 50.0%, indicating that MCP helps models improve their capabilities. On SLAKE, the gains are even more

| Dataset | Student Agent | # MCPs | Call Rate (%) |
|---------|---------------|--------|---------------|
| PathVQA | GPT-3.5-turbo | 9 | 38.0 |
| | Qwen3-8B | | 58.3 |
| | LLaMA3.1-8B | | 24.3 |
| SLAKE | GPT-3.5-turbo | 13 | 57.3 |
| | Qwen3-8B | | 94.7 |
| | LLaMA3.1-8B | | 57.0 |
| Game of 24 | GPT-3.5-turbo | 1 | 100 |
| | Qwen3-8B | | 100 |
| | LLaMA3.1-8B | | 100 |

Table 1: Generalizability and usage frequency of distilled MCPs across benchmarks. "# MCPs" denotes the number of distilled MCPs stored in the MCP Box, while "Calling Rate" indicates how frequently student agents invoke these MCPs.

pronounced—LLaMA3.1-8B by +10 points, GPT-3.5-turbo by +7.3 points and Qwen-8B improves by +6.7 points. On the arithmetic-focused Game of 24, GPT-3.5-turbo improves by 48.4 percentage points (34.3% to 82.7%), and LLaMA3.1-8B gains +42.3 points (21.7% to 64%). These consistent improvements across models and datasets demonstrate that MCP is effective in enhancing the task-solving ability of small language models (sLMs).

**Effectiveness across datasets.** AgentDistill yields consistent performance improvements across all datasets and base models. On SLAKE, all student models show notable gains, up to +10.0% for LLaMA3.1-8B, suggesting that semantically rich visual questions benefit from the compositional structure of distilled MCPs. Game of 24 exhibits especially large improvements for weaker models (e.g., +48.4% for GPT-3.5-turbo and +42.3% for LLaMA3.1-8B), indicating that MCPs effectively scaffold symbolic reasoning tasks such as arithmetic operations. In contrast, models that already perform well (e.g., Qwen3-8B on Game of 24) show smaller gains, likely due to ceiling effects(i.e., the strong baseline performance leaves limited room for improvement). Improvements on PathVQA are moderate but consistent, demonstrating the broad applicability of distilled MCPs.

**MCP Box narrows the gap between student agents and teacher agents.** To assess whether distilled MCPs help small language models (sLMs) approach the performance of much stronger agents, we compare MCP-equipped student agents with a reference teacher agent (Claude 4 + GPT-4o) and two retrieval-based systems: Octotools powered

| Dataset | Base Model | Before Distillation (%) | After Distillation (%) | Improvement (%) |
|---|---|---|---|---|
| PathVQA | GPT-3.5-turbo | 45.7 ± 3.5 | 52.7 ± 3.1 | +7.0 ↑ |
| | Qwen3-8B | 53.0 ± 1.7 | 55.3 ± 1.5 | +2.3 ↑ |
| | LLaMA3.1-8B | 46.7 ± 1.2 | 50.0 ± 1.7 | +3.3 ↑ |
| SLAKE | GPT-3.5-turbo | 61.0 ± 2.0 | 68.3 ± 0.5 | +7.3 ↑ |
| | Qwen3-8B | 61.0 ± 3.6 | 67.7 ± 2.1 | +6.7 ↑ |
| | LLaMA3.1-8B | 49.3 ± 2.9 | 59.3 ± 2.1 | +10.0 ↑ |
| Game of 24 | GPT-3.5-turbo | 34.3 ± 3.2 | 82.7 ± 0.6 | +48.4 ↑ |
| | Qwen3-8B | 72.7 ± 5.4 | 79.7 ± 6.1 | +7.0 ↑ |
| | LLaMA3.1-8B | 21.7 ± 4.7 | 64.0 ± 6.6 | +42.3 ↑ |

Table 2: Performance of student agents before and after distillation using **AgentDistill**. Accuracy improvements are observed across all datasets and models without any additional training.

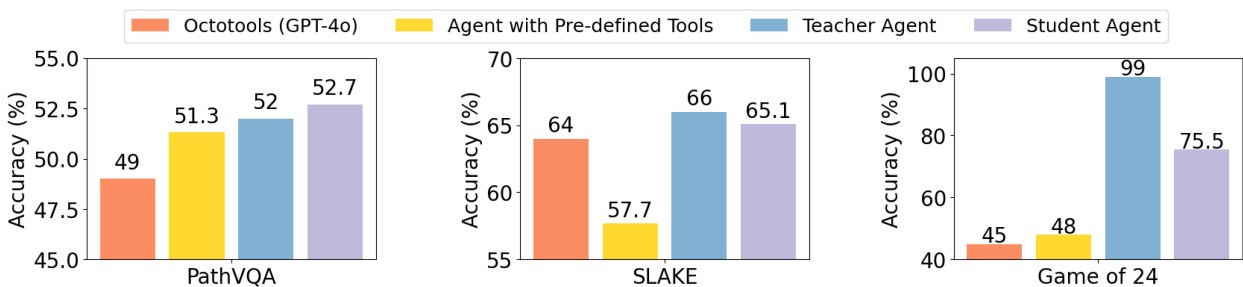

Figure 4: Comparison between the teacher agent (`Claude 4 + GPT-4o`) and the average performance of student agents (`GPT-3.5-turbo`, `Qwen-8B`, `LLaMA3.1-8B`) after distillations. The Octotools (`GPT-4o`) reports the performance of an open-source toolset baseline and the Agent with Pre-defined Tools (`GPT-3.5-turbo`, `Qwen-8B`, `LLaMA3.1-8B`) represents the average performance of sLM in Octotools with optimal toolsets. All agents operate without fine-tuning and student agents are evaluated with distilled MCPs.

by `GPT-4o`, and Agents with pre-defined tools upon Octotools Framework paired with sLMs, both equipped with optimal toolset (Figure 4). On PathVQA, average student agents after distillation (with the MCP Box) achieve 52.7% accuracy—matching the teacher agent (52%) and outperforming both retrieval-based variants. On SLAKE, MCP-equipped students reach 65.1%, slightly below the teacher (66%) but above both Octotools baselines. On Game of 24, while the teacher asignificantly outperforming Octotools with GPT-4o (45%) and also slightly surpassing Octotools with sLMs (48%). The latter is partly due to strong base performance of Qwen-8B on arithmetic tasks, which dominates the average within sLM-based Octotools. These results show that a well-curated, self-contained MCP Box enables small models to close the gap with much stronger agents, outperforming retrieval-based pipelines—even those backed by more powerful LLMs. This suggests that distilled MCP Box provides not only task transferability but also efficiency advantages over dynamic retrieval and tool orchestration.

### 4.3 ANALYSIS

**Why MCP Distillation works.** The MCP Box serves as an external library of executable protocols, distilled from teacher trajectories and abstracted for reuse. Each protocol encapsulates tool-level logic in a parameterized format, allowing the student agent to bypass low-level code generation. However, the student remains responsible for high-level planning: it must decide whether to invoke a tool, which MCP to select, and how to fill in the arguments. No policy gradients or planning heuristics are transferred; instead, the benefit arises from constraining the tool-calling space to a set of functional, verified options. This reduces generation complexity without interfering with the agent's core reasoning process.

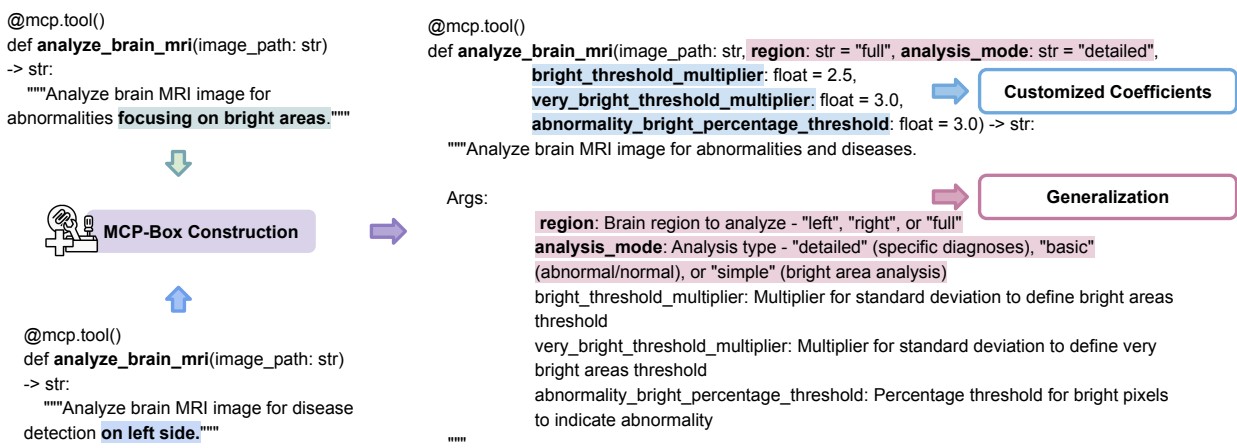

Figure 5: AgentDistill constructs a generalizable MCP from teacher-generated subtasks. MCPs on the left of this figure target specific goals (e.g., bright spot detection, left-side analysis), which are consolidated into a reusable parameterized MCP. The distilled MCP enables flexible reuse by adjusting arguments like `region` and `analysis_mode`, making it adaptable to different tasks without retraining.

**Case Study: Brain MRI Analysis** Fig. 5 highlights the core advantage of our AgentDistill framework: enabling student agents to acquire generalizable and reusable tools from teacher-generated protocols. In this example, the teacher produces two MCPs focused on narrow subtasks—detecting bright areas and analyzing the left hemisphere. AgentDistill then consolidates these into a parameterized MCP template that supports broader functionality. By exposing arguments like `region`, `analysis_mode`, and threshold multipliers, the distilled tool supports diverse configurations across brain regions, diagnostic modes, and image characteristics.

This design decouples task semantics from implementation logic, allowing the same MCP to be reused across new clinical scenarios (e.g., switching from MRI to CT, left-side to full-brain, simple detection to detailed diagnosis) without requiring additional code modifications. Such generalization is central to our training-free distillation pipeline, which converts ad-hoc language traces into structured, modular, and composable tools, ready to support student agents in dynamic or unfamiliar environments.

**Qualitative Ablation Discussion.** While our main results focus on demonstrating the overall effectiveness of the distilled MCP Box, we also analyze the necessity of each step in the construction pipeline. Without **abstraction**, MCPs remain tied to case-specific trajectories (e.g., an MRI MCP hard-coded for "left hemisphere" only) and cannot generalize to other inputs of the same property. Without **clustering and consolidation**, the MCP pool becomes bloated with near-duplicate tools that share similar functionality (e.g., multiple versions of MRI analysis with overlapping names and similar descriptions, as shown in Figure 5). Such redundancy potentially complicates tool invocation by increasing search space, token usage, and interface inconsistency.

## 5 CONCLUSION

We propose AgentDistill, a novel and training-free agent distillation framework that transfers task-solving capabilities from large teacher agents to small student agents through distilled Model–Context–Protocols (MCPs). Instead of relying on trajectory replay or gradient updates, the proposed method abstracts, clusters, and consolidates reusable tool-use strategies into an executable MCP Box, which is directly mounted into student agents at inference time. Experimental results on biomedical and mathematical benchmarks confirm that MCP-equipped student agents not only close the performance gap with teacher agents but also outperform retrieval-based systems like OctoTools(GPT-4o) even when using a strong LLM as the base model. The results highlight the potential of structured protocol distillation for enabling efficient, modular, and generalizable agent behavior without additional training or model modifications.

## REPRODUCIBILITY STATEMENT

All datasets used in this work (PathVQA, SLAKE, and Game of 24) are publicly available. The design of the MCP Box, including the abstraction, clustering, and consolidation steps, is described in Section 3. Details of the teacher and student agents, as well as baseline comparisons, are reported in Section 4.1 and Section 4.2. While we have not included full prompt templates in this submission, we plan to release them to ensure that all results can be faithfully reproduced.

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

## USE OF LLMS

LLMs were used solely to improve the clarity and readability of the manuscript.

