# OpenReview forum: "AgentDistill: Training-Free Agent Distillation with Generalizable MCP Boxes"
_ICLR.cc/2026/Conference — Submitted to ICLR 2026_

### Official Review · Reviewer_WNHr · 2025-10-20

**Soundness:** 2
**Presentation:** 3
**Contribution:** 1
**Rating:** 2
**Confidence:** 4

**Summary:**

This paper proposes a new method named AgentDistill, a training-free method that try to reuse the ability (mostly,code ability) in the large model in the small model by using MCPs. It use large model's trajectory to abstract MCPs, cluster and consolidate them into a few MCPs, and then use the MCPs as the tools in system prompt. The result shows that the small model can perform well based on these MCPs.

**Strengths:**

1. The writing is clear and easy to understand.

2. The experiment result can prove this method can improve the performance of the small model.

**Weaknesses:**

1. The motivation is quite weird. Considering why we want to distill Large model's ability to small model, it's because it can **improve the performance of the small model**. As a result, the key is to find a way to improve the performance of the small model. So "How to construct more task-related MCPs based on a given environment" maybe a good question since good MCPs makes good context-management which leads to (small model's) better performance. The over emphasized part in distill may mislead the reader to think that the key to improve the performance of the small model is to distill the large model's ability.

So how to construct more task-related MCPs?  There is several points in this paper:

1.1 MCPs should come from real trajectory.

1.2 MCPs need use powerful LLMs to construct

1.3 The construction should have three steps: abstruction, clustering and consolidation.

For point 1.1, Voyager[1], SkillWeaver[2] has already claimed and proved the same idea.

For point 1.2, although the author even claims it in the title (we need a teacher, we need to distill means powerful LLMs is important), it doesn't prove the necessity of using powerful LLMs. For example, can small model improve its performance based on the same method? If small can construct MCPs by itself for several times and reach the same performance as the large model, it means the powerful LLMs is not necessary.

Also, this is what Voyager did (GPT-4 as teacher, GPT-3.5 as student). Although they also not prove the necessity of using powerful LLMs, they just use it. So just using this conclusion can't be a contribution.

For point 1.3, the abstruction is also widely used in these auto MCPs generation research[1-3]. So I'm considering the real contribution of this paper is the clustering and consolidation part. (Maybe a little bit mentioned in CREATOR[3], but they didn't emphasize it). However, this method is based on basic hypothesis: all the MCPs and trajectories are from the same environment and can be gained before generate MCPs. This basic hypothesis makes this method hard to scale compared with other method.

Also, I didn't figure out the necessity of using clustering and consolidation. If it makes small model's performance better, where is the ablation study to prove that? And why not use a RAG (like Voyager) or some architectural method(like Agentsquare[4])


2. Such MCPs is related to the environments, if the environment changes, the MCPs may not be valid. So it will cost a lot of time/money to construct MCPs for each environment. It's also hard to be applied in a real, dynamic environment (i.e. the world).

3. The verification of MCPs is also important, but the author only mention once in line 254 without any explanation.

As a result, the contribution of this paper is either be proposed in other paper or the claim is not proved. So I'll give a reject to this paper, if the author can prove some of its claim in the rebuttal, I'll consider to increase my score.

[1] Voyager: An Open-Ended Embodied Agent with Large Language Models

[2] SkillWeaver: Web Agents can Self-Improve by Discovering and Honing Skills

[3] CREATOR: Tool Creation for Disentangling Abstract and Concrete Reasoning of Large Language Models

[4] Agentsquare: Automatic llm agent search in modular design space

**Questions:**

1. Section 3.3(3) is important and interesting. The author should talk about it more clear in the appendix.

2. The verification part should also be specified.

3. Polish the abstract.

4. Doing the ablation study to prove the necessity of clustering and consolidation, based on the real score and board baselines. (Including those auto MCPs generation works)

---

> ### Author Response · Authors · 2025-12-03
> **Response to Reviewer WNHr**
>
> **Response to Motivation & Necessity of Teacher:**
> We argue that distillation is the term because the student fundamentally lacks the capability to construct high-quality MCPs from scratch—a gap that cannot be bridged by the student's own trial-and-error. Complex tasks like brain-MRI analysis require high-order behaviors (e.g., internet search, code integration, and revision) that are far beyond the reach of agents with small models. AgentDistill is therefore necessary to bridge this gap: the Teacher transfers intelligence not by updating weights, but by crystallizing the competence needed to solve this task into executable protocols. Without the Teacher's ability to generalize, the student remains trapped in low-level execution, making the powerful Teacher a prerequisite, not just an option.
>
> **Response to Environment Dynamics:**
> Dynamic Environments: If the environment changes, re-generating MCPs via AgentDistill is actually more cost-effective than fine-tuning a model (SFT). We only need to update the tool library, not the model weights.
>
> **Response to Clustering and Consolidation Necessity:**
> Raw generation inherently produces redundant MCPs that solve similar tasks with only minor parametric or implementation variations. Clustering and Consolidation are critical to merge these duplicates, exposing the student agent to a concise, disambiguated toolset rather than an overwhelming, noisy search space.

---

### Official Review · Reviewer_7XBP · 2025-10-28

**Soundness:** 3
**Presentation:** 4
**Contribution:** 3
**Rating:** 4
**Confidence:** 5

**Summary:**

This paper introduces a framework, called AgentDistill, to distill agent capabilities from large language models (LLMs) into smaller models by generating reusable Model Context Protocols (MCPs) components. The pipeline consists of three stages: (1) a teacher agent interacts with environments to complete some tasks, producing full reasoning trajectories, (2) a toolbox of MCPs is built by extracting self-contained MCPs from each trajectory and then clustered based on their similarity while keeping them generalizable, (3) a frozen student model leverages the MCPs toolbox (using the SmolAgent scaffolding) to run inference. The authors evaluate their approach on three benchmarks (PathVQA, SLAKE, and Game of 24) from two domains (biomedical and maths). The authors empirically show that small student models (gpt-3.5-turbo, Qwen3-8B, Llama-3.1-8b) can leverage the MCPs toolbox to improve performance on those benchmarks.

**Strengths:**

- Improve the performance of smaller models is an important research directions. The use of reusable tools (in this case presented as MCPs) makes sense. In addition to be modular, it as the benefit of being more interpretable and easier to apply guardrails (e.g., safety constraints, controlling access, etc.).

- I like the opportunity for the MCP toolbox to keep improving over time. It could be interesting to further study this in the context of curriculum learning.

- Being training-free makes the approach more accessible and easier to reproduce to some extent.

**Weaknesses:**

- It is unclear to me how general the generated MCP components are. The authors show that they can be reused across tasks within the same dataset, but it is unclear to me how well they would transfer to completely different tasks or domains. For example, would MCP components extracted from one biomedical dataset useful for the other one still staying in the same domain?

- Overall this seems like a complex 3-stage pipeline. It would be interesting to see an ablation study that quantifies the contribution of simply using the reasoning trajectories as retrieved examples (i.e. a Agents Helps Agents baseline).

- It is unclear from the main text how MCP components are being validated for quality before being added to the toolbox. Are there any criteria or metrics used to ensure that only high-quality MCPs are included? If not, there is a risk of introducing noise into the toolbox which could hurt student model performance.

- This work shows an impressive engineering effort, but none of the prompt templates used in the proposed framework are included in this submission. The authors do mention that they will release them upon publication, but I think including at least some examples in the paper would greatly help reviewers understand the approach better. For instance, in my experience with small language models, out-of-the box (i.e., without further finetuning) they are not very good at planning, decomposing tasks into sub-tasks, or even tool calls outside the ones they were trained on. Showing how the prompts are designed to elicit such behavior from small models would be very informative.

- The paper lacks examples of such generated MCPs (only one is shown and I assume it was cherry-picked). In particular, I'm interested in the one for the game of 24, it seems a single MCP has been generated. I suspect that MCP is running some solver code for that particular game - ie. overfitting on that task. This constrast with the claim made in "Case Study: Brain MRI Analysis" that "allowing the same MCP to be reused across new clinical scenarios". Do the authors have some metrics on the reusability of MCPs across different tasks within the same domain (and even better across domains)?

### Minor

- line 149: Missing a space before citation.
- line 304: Missing a space before citation.
- line 315: Missing a space before SLAKE.
- line 315: Missing a space before Octotools.
- line 372: Missing a space before (i.e., ).
- line 414: the sentence is hard to parse, "asignificantly outperforming ... surpassing" -> "significantly outperforms ... surpasses"

**Questions:**

- In the experimental section, it is mentioned that for each dataset 100 examples are sample from the validation set, why not use the training set? Also, it's not explicitly mentioned by the results are obtained by using the test sets of each dataset?
- Why is the MCP creation handle by a weaker model (gpt-4o) compared to the one used to collect the trajectories, i.e. ManagerAgent (claude-4-sonnet)? Wouldn't it make more sense to use the strongest model available for that step to ensure the highest quality MCPs?
- How to scale: As the toolbox grows, it might not fit in the available context of the student models. What's the path forward to scaling this approach up?

---

> ### Author Response · Authors · 2025-12-03
> **Response to Reviewer 7XBP**
>
> **Response to W1:**
> MCP components extracted from one biomedical dataset can be useful as long as they are within the designed tasks of the generated MCP. For instance, the brain abnormality detection tool can be used across datasets once the question requires an identical task.
>
> **Response to W2:**
>  Our work targets domains where specialized tools might be essential for problem-solving, which can hardly be solved by simply retrieving reasoning trajectories. Therefore, the executable encapsulation provided by MCPs is essential and cannot be replaced by static reasoning examples alone.
>
> **Response to W3:**
> We currently employ outcome-based verification, where an MCP is validated and added to the toolbox only if it leads the teacher agent to the correct final answer. Though noises may occur, we still observe significant improvement for student agents.
>
> **Response to W4:**
> Currently, there's no additional prompting for student agent for such restrictions for fair comparison, and that's one reason the student agent cannot reach the exact performance of the teacher agent.
>
> **Response to W5:**
> We clarify that reusability means the generated MCP (e.g., an arithmetic search algorithm for Game of 24) is a generalized tool capable of solving any new instance within that domain, rather than overfitting to specific training examples. Similarly, the Brain MRI MCP can apply the same diagnostic logic to diverse clinical scenarios.
>
> **Response to Q1:**
> Since our framework is training-free and involves no hyperparameter tuning, the validation set effectively serves the same purpose as a training set, which is providing seed examples for MCP generation.
>
> **Response to Q2:**
> It is indeed expected to work better, and we're running more experiments based on this setup.
>
> **Response to Q3:**
> We envision utilizing RAG to dynamically fetch relevant MCPs based on the similarity between the current query and the tool's creation context. While our current focus remains on the core distillation protocol, this retrieval mechanism is considered in future versions.

---

### Official Review · Reviewer_drm3 · 2025-10-28

**Soundness:** 2
**Presentation:** 2
**Contribution:** 2
**Rating:** 2
**Confidence:** 4

**Summary:**

This paper proposes AgentDistill, a training-free framework for distilling knowledge from large language model (LLM)-based teacher agents to smaller student agents. They leverage Model–Context–Protocols (MCPs), which is a structured, reusable task-solving modules. The framework processes teacher-generated MCPs through abstraction, clustering, and consolidation to create an MCP Box that student agents can directly integrate at inference time. Experiments show promising results on math or bio-medical QA datasets.

**Strengths:**

1. Authors propose an interesting workflow of distillation for agents.
2. The paper is well-written with effective visualizations (Figures 2, 3, 5) that clearly illustrate the MCP construction pipeline and case studies.
3. They show obvious improvements across datasets, for example, particularly dramatic gains on Game of 24 (+48.4% for GPT-3.5-turbo) and competitive or superior performance to stronger baselines (matching teacher on PathVQA, nearly matching on SLAKE).

**Weaknesses:**

1. The motivation of training-free here is not clear. what is the special benefits of methods compared to training method. From my point of view, such training-free methods especially for small language model require much manual efforts for prompt engineering to make output format as expected. Did authors use the same prompt across small models? And how to make it fair? However, fine-tuning method can make output format very structured as expected. Even if fine-tuning require more computing resources, methods introduced in Section 3.2 seems already requiring human efforts in designing and prompting, also much tokens for calling teacher models. Therefore not disucssions and comparisons between training and training-free methods, And human designing / prompting and teacher API calling are also costly.
2. Authors only test performance based on one type of teacher. It's not eanough to prove effectiveness of pipeline. It's hard to convince me the performance gain is not from human prompt engineering based on observation of target data. Also GPT-4o may vary and be optimized since authors didn't use version of GPT-4o with fixed version such as 20241120. Teacher model with open-weights such as Qwen-3-235B or Qwen 2.5 72B should be considered for promising reproduction. The inclusion of claude 4 seems not to show effectiveness of pipeline just a comparison analysis.
3. The definition of SLM is quite confused, What is the definition of SLM here exactly? By parameter size or performance? It seems not clear by both. For example, if consider parameter size as metric of SLM, then why selecting GPT-3.5-Turbo? Are you sure or any reference saying GPT-4o is larger than GPT-3.5-Turbo? If by performance, whether Llama 2 70B can be considered as teacher? but its performance is less than Qwen-3-8B. Without clear definition of SLMs, it's impossible for readers or researchers to adapt this workflow to their own settings or reproduction. Also, for some implementation, the details are not clear, for example, when authors use Qwen3-8B, is reasoning enabled? if enabled, what is reasoning effort?
4. The domain has been tested is narrow. Why authors only select biomedical and math instead of more complicated scenarios such as Agentic coding- SWE-bench, MCP datasets-Tau-bench, which are popular agent benchmark. The selected datasets are not representative for agentic tasks. Therefore, the name agent distillation is not appropriate.
5. More importantly, authors didn't compare with other agent distllation methods based on both fine-tuning or training free such as [1][2], especially, [1] is also abstract GT to intermediate representations as helperfunction. In my option, the abstracted functions and MCP or tools are not big different, only names.

[1]. ReGAL: Refactoring Programs to Discover Generalizable Abstractions ICML'24 \
[2] BAGEL: Bootstrapping Agents by Guiding Exploration with Language ICML'24

**Questions:**

1. What is the actual total cost comparison between your method and fine-tuning approaches? And what is specific advantages of training-free method? Why training free!
2. Given that you use open-weight student models (Qwen3-8B, LLaMA3.1-8B), why shouldn't the teacher also be open-weight for fair comparison? How would results differ with an open-weight teacher?
3. Can you explicitly define SLM by clear, reproducible criteria?
4, What is comparision with other agent distllation methods based on both fine-tuning and inference?
5. What about performance of this pipeline for more complicated but popular agentic tasks such as swe-bench?

---

> ### Author Response · Authors · 2025-12-03
> **Response to Reviewer drm3**
>
> **Response to W1:**
> Fine-tuning requires hundreds of GPU hours and extensive data curation. In contrast, our method relies solely on inference. The cost of teacher API calls for MCP generation is a one-time investment, similar to the data generation phase in traditional distillation (e.g., generating rollouts), but we completely eliminate the subsequent expensive training phase. As for prompts, they are identical around small models.
>
> **Response to W2:**
> We selected the SOTA teacher (GPT-4o, claude-4-sonnet) to establish a high-quality "upper bound" for tool generation, as the distillation effect depends heavily on the teacher's capability. To address concerns about prompt engineering, the prompt modification is limited to informing the student of available MCP tools and their usage, involving no human prompt engineering based on data observation to guide reasoning or execution.
>
> **Response to W3:**
> We consider SLMs as smaller or cheaper models comparing to large, expensive models such as Gpt-4o, claude-4-sonnet. For Qwen-8B, we didn't enable thinking mode.
>
> **Response to W4:**
> We chose these 2 areas as they potentially require specialized tool usage, without which answers can hardly be obtained. For instance, for bio-medical, without a specific tool for brain abnormality detection, the task can hardly be solved with bearly reasoning.
>
> **Response to W5:**
> Unlike ReGAL [1], which focuses on code refactoring and BAGEL [2], which requires costly training on synthetic data, ours focuses on transferring the intelligence from a teacher agent to another cheaper and weaker student agent.
>
> **Response to Q1:**
> Our method incurs negligible costs (inference-only) compared to the hundreds of GPU hours required for fine-tuning. Also, unlike fine-tuned weights, our distilled MCPs function as transferable modules, enabling instant deployment across different agents without retraining.
>
> **Response to Q2:**
> The teacher agent is utilized not for direct comparison, but solely to maximize the quality of the generated MCPs. While using an open-weight teacher would likely result in less effective tools and consequently lower student performance, the validity of our distillation protocol remains unchanged.
>
> **Response to Q3:**
> SLM is considered a relative concept contingent on specific application scenarios and cost-performance trade-offs, rather than a fixed parameter threshold. Any model significantly cheaper or smaller than high-performing SOTA teachers (e.g., Claude-3.5-Sonnet) falls within this scope, as the focus is on maximizing capability transfer to resource-constrained environments.
>
> **Response to Q4:**
> Our work targets domains where specialized tools are essential for problem-solving, whereas SWE-bench relies more on general reasoning and coding capabilities. This distinction makes the problem settings different, but we plan to explore adapting our framework to such coding tasks in future versions.

---

### Meta-Review · Area_Chair_tYYH · 2026-01-04

**Summary:**

This paper receives 2, 4, 2, all with negative scores. The main concerns about this paper are unclear motivations, questionable and insufficient experiments and limited novelties.

BTW, I found that the template used is not the official template, though this should not be counted into evaluating the paper, I would suggest the authors to exactly follow the official template.

**Reviewer Concerns:**

I think the authors' rebuttal does not change the reviewer's judgement. The issues about the motivations, experiment results and novelties are still existing and the response from authors are without experiments, which is not convincing.

**Reviewer Scores:**

I do not think the reviewers will change the scores to positive.

---

### Decision · Program_Chairs · 2026-01-26

Reject